# Engineering a Dual Specificity γδ T-Cell Receptor for Cancer Immunotherapy

**DOI:** 10.3390/biology13030196

**Published:** 2024-03-20

**Authors:** David M. Davies, Giuseppe Pugliese, Ana C. Parente Pereira, Lynsey M. Whilding, Daniel Larcombe-Young, John Maher

**Affiliations:** 1Leucid Bio Ltd., Guy’s Hospital, Great Maze Pond, London SE1 9RT, UK; mdavies@leucid.com (D.M.D.); giuseppe.pugliese.med@gmail.com (G.P.); 2Department of Oncology and Hematology, University Hospital of Modena, 41124 Modena, Italy; 3CAR Mechanics Group, Guy’s Cancer Centre, School of Cancer and Pharmaceutical Sciences, King’s College London, Great Maze Pond, London SE1 9RT, UK; anacatpp@gmail.com (A.C.P.P.); lynseywhilding@hotmail.co.uk (L.M.W.); daniel.larcombe-young@kcl.ac.uk (D.L.-Y.); 4Department of Immunology, Eastbourne Hospital, Kings Drive, Eastbourne BN21 2UD, UK

**Keywords:** γδ T-cell, Vγ9Vδ2 receptor, αvβ6 integrin, cancer, A20 peptide, foot and mouth disease virus

## Abstract

**Simple Summary:**

A protein known as the γδ T-cell receptor is the distinguishing hallmark of a small white blood cell subset known as γδ T-cells. γδ T-cells survey the body for potentially dangerous cells, including cancer. The most common γδ T-cell receptor found in the bloodstream is called Vγ9Vδ2 and this detects attributes of cancer cells called phosphoantigens. To generate large numbers of cells that detect phosphoantigens, the Vγ9Vδ2 T-cell receptor gene can be introduced into a more plentiful white blood cell subset called αβ T-cells. In this study, we have inserted a small additional peptide fragment into the Vγ9Vδ2 T-cell receptor, enabling it to also recognise a second tumour target called αvβ6 integrin. When this modified receptor was expressed in αβ T-cells, they acquired the ability to kill tumour cells that produce either phosphoantigens or αvβ6 integrin. By this means, we have broadened the tumour recognition capacity of these engineered T-cells.

**Abstract:**

γδ T-cells provide immune surveillance against cancer, straddling both innate and adaptive immunity. G115 is a clonal γδ T-cell receptor (TCR) of the Vγ9Vδ2 subtype which can confer responsiveness to phosphoantigens (PAgs) when genetically introduced into conventional αβ T-cells. Cancer immunotherapy using γδ TCR-engineered T-cells is currently under clinical evaluation. In this study, we sought to broaden the cancer specificity of the G115 γδ TCR by insertion of a tumour-binding peptide into the complementarity-determining region (CDR) three regions of the TCR δ2 chain. Peptides were selected from the foot and mouth disease virus A20 peptide which binds with high affinity and selectivity to αvβ6, an epithelial-selective integrin that is expressed by a range of solid tumours. Insertion of an A20-derived 12mer peptide achieved the best results, enabling the resulting G115 + A12 T-cells to kill both PAg and αvβ6-expressing tumour cells. Cytolytic activity of G115 + A12 T-cells against PAg-presenting K562 target cells was enhanced compared to G115 control cells, in keeping with the critical role of CDR3 δ2 length for optimal PAg recognition. Activation was accompanied by interferon (IFN)-γ release in the presence of either target antigen, providing a novel dual-specificity approach for cancer immunotherapy.

## 1. Introduction

γδ T-cells account for up to 10% of circulating lymphocytes and operate at the interface between innate and adaptive immunity. These cells also provide an important line of defence against malignant transformation [1]. Indeed, the presence of intra-tumoural Vγ9Vδ2 T-cells is the most predictive leukocyte signature of improved outcomes across 25 cancer types [2]. 

To mediate cancer immunosurveillance, γδ T-cells recognise genomic, metabolic, and signalling perturbations associated with the transformed state, coupling this to the triggering of T-cell effector function [3]. They also act as efficient antigen-presenting cells, enabling the perpetuation of immune attacks through adaptive mechanisms [4]. Since γδ T-cells are not HLA-restricted, they do not elicit graft versus host disease. Consequently, these cells are of potential utility for the allogeneic or “off the shelf” treatment of cancer. 

Most circulating γδ T-cells in man display a Vγ9Vδ2 receptor that recognises non-peptide phosphoantigens (PAgs) [5] in a butyrophilin 3A1 and butyrophilin 2A1-dependent manner [6]. Since PAgs are intermediates of mevalonate metabolism, Vγ9Vδ2 T-cells provide an innate mechanism to detect overactivity of this key metabolic pathway. Such surveillance is justified since excess mevalonate pathway flux contributes importantly towards cellular transformation [7]. 

Despite these considerations, immunotherapy using γδ T-cells has achieved limited efficacy thus far [8]. One strategy that has been used in an effort to overcome this entails the genetic delivery of Vγ9 and Vδ2 TCR genes into the more plentiful αβ T-cell population [9]. This approach was first exemplified using the clonal G115 γδ TCR [9] and, more recently, using higher affinity Vγ9Vδ2 TCRs [10]. While some clinical efficacy has been achieved with such γδ TCR-engineered T-cells, potentiation of this system has recently been proposed via the co-expression of a chimeric co-stimulatory receptor in these cells [11].

In this study, we sought an alternative method to enhance the therapeutic efficacy of γδ TCR-engineered T-cells by broadening the intrinsic tumour specificity of the TCR itself. Using the G115 Vγ9Vδ2 TCR, it has been shown that amino acids L109 and T113 onwards within the Vδ2 complementarity determining region (CDR) 3 region are all critical for TCR expression and/or function. Figure 1A highlights these key residues within the G115 TCR. Surprisingly, however, amino acids between L109 and T113 could be replaced by as many as 11 alanine residues without disrupting TCR-mediated PAg recognition [12]. This finding raises the possibility that a short tumour-specific peptide could be substituted for residues at this location, thereby generating a dual specificity TCR. To test this, we selected the A20 foot and mouth disease virus (FMDV) peptide [13], which binds to the αvβ6 integrin. αvβ6 integrin is commonly over-expressed by a wide range of epithelial tumours, with minimal expression in healthy epithelium [14]. The A20 peptide has previously been used to confer tumour specificity on chimeric antigen receptor (CAR) T-cells [15,16,17] and a carcinoembryonic antigen-specific single-chain antibody fragment, via CDR3 grafting [18]. In that case, a 17 mer peptide proved sufficient to confer αvβ6 integrin reactivity. Here, we evaluated whether a similar CDR3 grafting strategy could be utilised to confer specificity for αvβ6 integrin upon γδ TCR-engineered T-cells.

## 2. Materials and Methods

### 2.1. Cell Lines

All cell lines used in this study were validated by STR typing and were confirmed as mycoplasma negative. The K562 and Jurkat E6.1 cell lines were provided by Dr Linda Barber, King’s College London, UK. The MDA-MB-468 breast cancer cell line was provided by the Breast Cancer Now Research Unit, King’s College London, UK. Pancreatic cancer cell lines (BxPC3 and Panc1) and isogenic A375 puro and A375-β6 cells were provided by Prof John Marshall, Queen Mary University of London, UK. To monitor tumour cell viability using luciferase assays, firefly luciferase (ffLuc) was expressed in target cells by retroviral transduction, as described [15]. Human Embryonic Kidney (HEK)293T cells were obtained from the American Tissue Culture Collection. All cells were maintained in DMEM medium that contained 10% FBS and GlutaMax (D10 medium).

### 2.2. Human Samples

Blood samples were collected from healthy male and female volunteers, aged between 18–65 years under the authority granted by a National Health Service Research Ethics Committee (code 09/H0804/92 and 18-WS-0047).

### 2.3. Retroviral Constructs

To construct the SFG G115 vector, a codon-optimised sequence encoding the G115 γ9 chain, T2A peptide sequence, and G115 δ2 chain were synthesized by Integrated DNA Technologies (IDT; Coralville, IA, USA). This fragment also contained the upstream bases present between the Age I and Nco I sites within the SFG retroviral vector and was ligated into the unique Age I and Xho I restriction sites in the SFG vector plasmid (all restriction endonucleases were from New England Biolabs, Hitchen, UK). G115 amino acid sequences were taken from [20]. To generate a construct that only encodes for the G115 δ2 chain alone, a δ2 cDNA fragment, including the upstream bases present between the Age I and Nco I sites within the SFG retroviral vector, was synthesized by IDT and ligated into the unique Age I and Xho I restriction sites in SFG. 

To generate TCRs in which A20-derived peptides shown in Figure 1A were inserted into G115 δ2 CDR3, codon-optimised DNA fragments were synthesised by IDT. These sequences were flanked at the 5′ and 3′ ends by codon-optimised DNA sequences from the endogenous Vδ2 variable region (bases 319–348) and Vδ2 joining region (bases 367–399), respectively. Each fragment was cloned into the SFG G115 construct using BamH I and Mlu I. In each case, the inserted peptide replaced the indicated endogenous Vδ2 sequence, thus giving SFG G115 + A7, SFG G115 + A12, SFG G115 + A20, and SFG G115 + A20 ΔCDR3 (Figure 1A).

### 2.4. Production of Retroviral Vector

Vector production was performed as described [21]. In brief, 3 × 10^6^ HEK293T cells were plated into 10 cm^2^ tissue culture-treated plates. Twenty-four hours later the cells were transfected with Genejuice (Merck Millipore, Burlington, MA) miscelles containing plasmids encoding the *gag*, *pro*, *pol*, and *env* retroviral genes alongside SFG encoding for the TCR of interest. Supernatants were harvested at both 48 and 72 h post-transfection and snap-frozen in an ethanol bath. Where T-cells were transduced with a mixture of two viral vectors (e.g., G115 + A20ΔCDR3 and Vδ2 alone vector), equal volumes of each vector were added at all incubations.

### 2.5. T-Cell Activation

Peripheral blood mononuclear cells (PBMC) were isolated using ficoll-based density gradient centrifugation as described [21]. Isolated cells were cultured in RPMI media supplemented with 2 mM L-glutamine and 5% human AB serum. T-cells were activated by the addition of CD3/CD28 Dynabeads (ThermoFisher Scientific, Waltham, MA, USA) at a 1:1 bead-to-cell ratio for 48 h. 

### 2.6. T-Cell Transduction

Non-tissue culture treated 6 well plates were pre-coated at 4 °C overnight with 200 µg RetroNectin (Takara, Shiga, Japan) per plate. Following the removal of the RetroNectin, the retroviral vector (3 mL per well) was pre-loaded onto the plate at 4 °C overnight. After removal of the vector, 1 × 10^6^ activated PBMC were added to each well followed by 3 mL additional retroviral supernatant plus 100 IU/mL rhIL-2 (Clinigen, London, UK). The cells were incubated for 10 days at 37 °C + 5% CO_2_ and were fed with 100 IU/mL rhIL-2 every 48 h. When necessary, the cells were split 1:2 and fed with fresh medium and rhIL-2. Prior to co-culture with target cells, the percentage of transduced T-cells was rendered equivalent between the various constructs by dilution with untransduced T-cells from the same donor. 

### 2.7. FACS Analysis

Flow cytometry was undertaken with a FACSCanto II or LSR Fortessa (BD Biosciences, Winnersh Triangle, UK) using FACSDiva software version 6.1.3 (BD Biosciences), recording at least 5 × 10^5^ events. Compensation settings were established using single-stained samples. Cells were first gated based on forward (FSC-A) and side (SSC-A) scatter (measuring cell size and granularity respectively) to exclude debris. Single cells were then selected using SSC-H versus SSC-W parameters. Dead cells were excluded using a viability stain.

All incubations were performed on ice. To detect γδ TCR expression, FITC conjugated pan γδ antibody (BD Biosciences), PE-conjugated pan γδ antibody (eBioscience, Altrincham, UK), or FITC conjugated δ2-specific antibody (Biolegend, San Diego, CA, USA) were used. Integrin expression was detected by flow cytometry using the following antibodies: αvβ3 integrin (FAB3050A, R&D Systems, Minneapolis, MN, USA), αvβ5 integrin (FAB2528A, R&D systems), αvβ6 integrin (FAB4155A, R&D Systems) and αvβ8 integrin (FAB4775A, R&D systems). Detection of the A20 peptide within engineered G115 TCRs was detected using the F21-64 monoclonal antibody [22]. Integrin binding by T-cells was determined by incubation with 1000 ng of His tagged human αvβ3, αvβ5, αvβ6 or αvβ8 integrin (ACROBiosystems, Newark, DE, USA). After washing in PBS, binding was detected using 2 μL of anti-His tag PE (R&D Systems, Minneapolis, MN, USA). All data were analysed using FlowJo version 9 (FlowJo, Ashland, OR, USA).

### 2.8. Stimulation of Engineered T-Cells on Immobilised αvβ6 Integrin

Recombinant αvβ6 integrin (3871-AV, R&D Systems) was resuspended at a final concentration of 800 ng/mL in sterile PBS containing calcium and magnesium and 100 µL/well added to non-tissue culture 24 well plates. Following incubation of the plates at 4 °C overnight, the PBS was gently removed and 1 × 10^5^ cells expressing the G115-derived TCR variants were added to the well in 1 mL of R5 medium. 

### 2.9. Enzyme-Linked Immunosorbent Assay

Media that were collected from T-cell/tumour cell co-cultures were analysed for interferon (IFN)-γ using ELISA (eBioscience), as described by the manufacturers.

### 2.10. Cytotoxicity Assays

Co-cultivation assays between T-cells and target cells were established at effector-to-target (E:T) ratios as specified in individual experiments. Target cell destruction was quantified by in vitro MTT or luciferase assays, as described in [21]. Using either assay, the percentage of target cell viability was calculated using the following formula: (absorbance/luminescence of tumour cells cultured with T cells divided by absorbance/luminescence of tumour cells alone) × 100%. Where indicated, tumour cells were pre-sensitized with 1 μg/mL zoledronic acid (Zometa, Novartis, East Hanover, NJ, USA) for 24 h prior to initiation of co-cultures.

### 2.11. Statistical Analysis

The normality of all experimental data was tested using the Shapiro-Wilk test prior to statistical analysis. Statistical analysis was performed using one-way or two-way ANOVA. All statistical testing was undertaken using GraphPad Prism (version 10.1.1, GraphPad Software, San Diego, CA, USA). 

## 3. Results

### 3.1. Engineering of CDR3 Mutant Derivatives of the G115 γδ TCR

To confer PAg specificity on conventional αβ T-cells, CD3+CD28 Dynabead-activated PBMC were transduced with an SFG retroviral vector that encodes for a Vγ9Vδ2 TCR known as G115 [9]. Stoichiometric transgene expression was achieved by placing a Thosea Asigna (T2A) ribosomal skip peptide between the γ and δ chain encoding cDNA sequences (Figure 1B). In an attempt to confer specificity for αvβ6 integrin upon this TCR complex, derivatives of the A20 FMDV 20mer peptide were inserted between the L109 and T113 residues within the Vδ2 CDR3 region (Figure 1A), removing intervening amino acids at that location and giving rise to the predicted γδ TCR structure shown in Figure 1C. Since previous studies have indicated that the length of the Vδ2 CDR3 region is critical for TCR function, we evaluated the insertion of a 7mer, 12mer, and full-length 20mer at that site, giving G115 + A7, G115 + A12, and G115 + A20 respectively (Figure 1B). Peptide selection was influenced by the fact that A20 contains two overlapping αvβ6-binding motifs (RGD and DLXXL), both of which were fully preserved in all three peptide variants [23]. Given that the largest peptide insert previously incorporated into the G115 Vδ2 CDR3 region contained only 11 amino acids [12], we also designed a further variant in which all residues within Vδ2 CDR3 region were removed prior to insertion of the intact A20 peptide (G115 + A20 ΔCDR3). 

To demonstrate cell surface expression of G115 and these A20-modified derivatives in transduced T-cells, flow cytometry was performed using a pan γδ TCR antibody. Figure 2 demonstrates that transduction efficiency was not hindered by the incorporation of either A20, A12, or A7 peptides provided that amino acids L109 and T113 onwards within the Vδ2 CDR3 region were maintained. However, the substitution of all Vδ2 CDR3 residues by the full-length 20mer peptide (G115 + A20 ΔCDR3) prevented cell surface expression of this chimeric γδ TCR in a manner that was complemented by ectopic expression of a wild type δ2 G115 TCR sequence (Appendix A). In the case of G115 + A20 but not G115 + A12, cell surface expression of the chimeric γδ TCR could also be detected in both transduced Jurkat cells and human T-cells with an A20-specific monoclonal antibody (Appendix A). Expression of chimeric γδ TCRs was detected at similar levels using either a pan-γδ TCR or δ2 chain-specific TCR antibody (Appendix A). Owing to competition for endogenous CD3 subunits, expression of the endogenous αβ TCR was reduced, but not abrogated, in the transduced T-cells (Appendix A). This finding is consistent with previous results [12]. 

### 3.2. Evaluation of αvβ6 Integrin Specificity of Engineered G115 γδ TCRs

Co-cultivation assays were conducted to determine the ability of the peptide-engineered G115 TCRs described above to detect αvβ6 integrin. First, T-cells were incubated with A375 cells which naturally express several RGD-binding integrins, including αvβ3, αvβ5, αvβ8 (low level), but not αvβ6 (A375 puro), making comparison with an isogenic β6-engineered derivative (A375-β6; Figure 3A) [15]. 

A375 cells are naturally resistant to killing by G115-engineered T-cells (Appendix A) allowing us to assess the ability of chimeric G115 TCR derivatives to trigger a cytolytic response in an αvβ6 integrin-dependent manner. As expected, we observed that G115-transduced T-cells exerted no cytotoxic activity against either A375 puro (Figure 3B) or A375-β6 cells (Figure 3C). By contrast, the insertion of an A12 FMDV peptide into Vδ2 CDR3 within this TCR enabled the selective cytolytic destruction of A375-β6 cells (Figure 3B,C). No significant tumour cell killing was observed with the G115 + A7 derivative (Figure 3B,C), as a result of which the further testing of this construct was terminated. The introduction of the full A20 peptide (G115 + A20) induced a less convincing cytolytic response against A375-β6 target cells (Figure 3B,C), suggesting that this peptide may be too long for optimal folding at this site. 

In the case of G115 + A12 cells, cytolytic activity was accompanied by an increase in interferon (IFN)-γ production (Figure 3D). Moreover, G115 + A12 cells demonstrated the selective ability to bind to αvβ6, but not to other integrins tested (Appendix A). Together, these data indicate that incorporation of A12 into the G115 TCR enables it to effectively detect the presence of αvβ6 integrin. 

### 3.3. Evaluation of PAg Specificity of Engineered G115 γδ TCRs on K562 Cells

Next, we evaluated whether the introduction of the A12 or A20 FMDV peptides into the G115 TCR had altered the ability of this TCR to recognise PAgs. For these studies ffLuc-expressing K562 cells were selected since they lack expression of the epithelial-specific αvβ6 integrin (Figure 4A) and present PAgs to γδ T-cells following treatment with the aminobisphosphonate, zoledronic acid (Zol) [24]. Figure 4B demonstrates that T-cells engineered with G115 alone or that incorporate either the A12 or A20 peptide achieve Zol-dependent cytotoxicity against ffLuc^+^ K562 cells. Surprisingly, tumour cell killing by G115 + A12 cells was significantly enhanced compared to G115 cells, most noticeably at lower concentrations of Zol. This confirms that the inclusion of either A12 or A20 does not compromise PAg recognition and suggests that this process may be enhanced in the case of the A12 construct. Once again, cytotoxicity was accompanied by the production of IFN-γ by G115 + A12 T-cells (Figure 4C).

### 3.4. Evaluation of Anti-Tumour Activity of Engineered G115 γδ TCRs on Pancreatic Tumour Cells

To test the generality of these findings, we next undertook further co-cultivation studies of G115 + A12 and G115 + A20-engineered T-cells with additional tumour cell lines. Pancreatic tumour cells were selected which do (BxPC3, Figure 5A) or do not (Panc1, Figure 6A) express αvβ6 integrin. To assess PAg recognition, co-cultures were conducted using Zol-sensitised or unsensitised target cells. In the case of unsensitised BxPC3 tumour target cells, G115 + A12 T-cells exerted significantly greater cytolytic activity than G115 T-cells alone, consistent with αvβ6 engagement (Figure 5B,C and Appendix A). Although the cytolytic activity of G115 T-cells was further enhanced when monolayers were pulsed with Zol, this was not the case for the G115 + A12 TCR. Similar findings were obtained using αvβ6^+^ MDA-MB-468 triple-negative breast cancer cells (Appendix A), once again consistent with recognition of αvβ6 integrin. By contrast, αvβ6 negative Panc1 cells were only killed when pre-sensitised with Zol (Figure 6B,C), indicative of PAg recognition. To further confirm the specificity of these chimeric TCRs for αvβ6 integrin, engineered T-cells were stimulated for 72 h on immobilised αvβ6, making a comparison with PBS. This resulted in a significant increase in IFN-γ production by G115 + A12 T-cells compared to G115 or untransduced controls (Appendix A). Taken together, these data reinforce the evidence that the G115 + A12 is a dual specificity TCR. 

## 4. Discussion

Naturally occurring dual reactivity of both human and mouse γδ TCRs has been described previously. In this context, some receptors combine an innate butyrophilin-dependent recognition capacity, with an adaptive-like response to additional targets [25]. Additionally, γδ T-cells that react with CD1b-presented lipid and butyrophilin-like proteins have been described [26]. While the innate immune recognition properties of γδ T-cells are well recognised, their ability to participate in adaptive immune responses is illustrated by their expansion in infections mediated by several herpes virus family members [26]. Moreover, traditional αβ TCRs with reactivity for a plurality of tumour antigens have recently been described in TIL (tumour-infiltrating lymphocyte) cell products associated with successful tumour control [27]. Taken together, these considerations indicate that poly-reactivity may represent an important intrinsic attribute in the selection of clinically useful tumour-specific TCRs. 

In this study, we have demonstrated the novel concept that cross-reactivity may be engineered into a γδ TCR of the Vγ9Vδ2 subtype using a synthetic biology-based approach. We demonstrate that the flexible nature of CDR3 length within the δ2 TCR subunit may be exploited not only to improve the recognition of PAgs but also to enhance tumour recognition through the inclusion of a distinct tumour-binding peptide. To our knowledge, this is the first demonstration that a TCR of defined antigenic reactivity can be modified to acquire a second specificity for antigen. An αvβ6 integrin-specific A20 peptide and shortened derivatives were inserted into the δ2 CDR3 region of the G115 receptor. This approach was selected since the key overlapping αvβ6-binding motifs (RGD and DLXXL) within this peptide are contained within a short, mobile, and self-contained loop. Consequently, we anticipated that this sequence would be accessible to αvβ6 integrin if inserted into the δ2 CDR3 region [28]. We took advantage of the fact that 6 amino acids within this region (residues 110, 111, 111A, 112, 112A, and 112B) are redundant for the PAg reactivity of this receptor. By replacing these 6 residues with a 12mer derived from the A20 FMDV protein, selective reactivity to αvβ6 integrin was conferred onto this receptor. Although the A20 FMDV peptide forms a complex helical structure [13], which could have perturbed the correct folding of the G115 TCR, PAg reactivity of the resultant G115 + A12 receptor was fully maintained, confirming that it is truly a dual specificity TCR. In keeping with this, G115 + A12-engineered T-cells were capable of killing tumour cells that express αvβ6 integrin alone, PAg alone, or both of these targets. Indeed, the killing of K562 target cells by G115 + A12 T-cells was increased compared to G115 controls when pulsed with low concentrations of Zol. This suggests that when PAg concentrations are limiting, the sensitivity of the receptor is increased, perhaps because of the incorporation of a more optimal peptide length at this site. In keeping with this, prior studies have indicated that the length of the Vδ2 CDR3 region within the G115 receptor is a key determinant of PAg responsiveness [12]. Although the full A20 peptide has previously been used to re-target CAR T-cell specificity, it proved less efficient in the context of a chimeric TCR (G115 + A20). This suggests that testing of additional A20-derived peptide lengths spanning the interval between A7 (poorly effective) and A20 (sub-optimally effective) could yield chimeric TCRs with greater reactivity against one or both targets. Reasoning that the poorer performance of the G115 + A20 TCR was due to the excessive size of the A20 peptide, further CDR3 residues were removed from the G115 δ2 CDR3 region but this prevented cell surface expression of the resulting G115 + A20 ΔCDR3 chimera. This was confirmed by co-transduction with a second vector encoding for a wild type δ2 chain and which could effectively restore cell surface expression of the G115 TCR by pairing with the Vγ9 chain incorporated in the G115 + A20 ΔCDR3 vector.

In this manuscript, we have exemplified this approach in αβ T-cells, introducing both the Vγ9 and modified Vδ2 TCR subunit using a single vector. In principle, the approach could also be used in circulating γδ T-cells, leading to the generation of a mixture of modified and unmodified TCRs in the same cell type. In both cases, competition between endogenous and exogenous TCR for CD3 subunits and downstream signalling partners would be expected to occur. Consequently, it may be desirable to engineer endogenous TCR subunits in order to maximise cell surface expression and signalling by the introduced chimeric TCR [29,30].

## 5. Conclusions

This study provides proof of concept that Vγ9Vδ2 TCRs can be engineered to have specificity against a carefully selected tumour-associated antigen, whilst maintaining endogenous recognition of PAgs. This dual specificity may reduce the impact of common tumour escape mechanisms, such as antigen loss and heterogeneity [31,32], and expands upon the growing use of such approaches to overcome tumour resistance, including OR-gate CARs, or the use of CARs that target multiple antigens simultaneously [33].

Impact of the proposed approach on in vivo anti-tumour activity remains to be determined and warrants further study. Furthermore, studies of downstream signalling pathways triggered by either PAg or αvβ6 integrin stimulation could provide interesting insights into the functionality of this chimeric TCR. Our study raises the possibility that additional dual specificity receptors could be generated by the insertion of alternative tumour-specific peptides at this location. 

## Figures and Tables

**Figure 1 biology-13-00196-f001:**
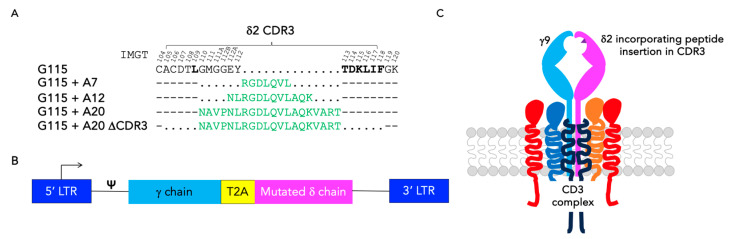
Engineering a chimeric G115 γδ T-cell receptor. (**A**) The sequence of the δ2 complementarity determining region (CDR); three regions of the G115 γδ TCR are shown, with numbering according to the international ImMunoGeneTics information (IMGT) system [19]. Residues shown in bold are required for expression and/or phosphoantigen recognition by this TCR. The A20 FMDV-derived 7mer, 12mer, and 20mer peptides which were inserted into this CDR3 region are shown in green text, together with the nomenclature of G115-derived TCRs. Conserved amino acids are indicated by a dash while absent amino acids (G115) or those removed from the consensus G115 sequence (all other TCRs) are indicated by a dot. In G115 + A20ΔCDR3, all residues within the δ2 CDR3 region have been removed and the full-length A20 peptide was inserted instead. (**B**) Structure of the SFG retroviral vector used to deliver G115 and derived γδ TCRs to human T-cells. LTR–long terminal repeat; T2A–*Thosea Asigna* ribosomal skip peptide. (**C**) Predicted cartoon structure of derived G115 TCRs.

**Figure 2 biology-13-00196-f002:**
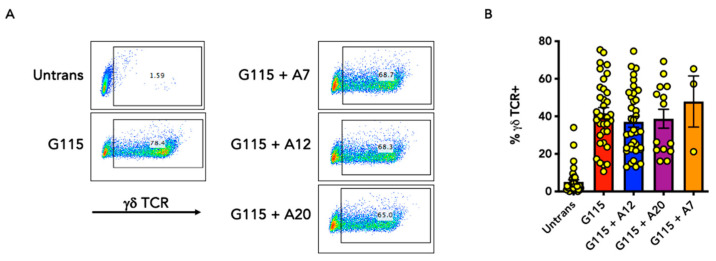
Expression of chimeric G115 γδ TCRs in human T-cells. (**A**) Representative flow cytometry plots that demonstrate cell surface γδ TCR expression in Dynabead-activated T-cells following transduction with the indicated construct. (**B**) Pooled data from individual donors (n = 3–36, mean ± SEM).

**Figure 3 biology-13-00196-f003:**
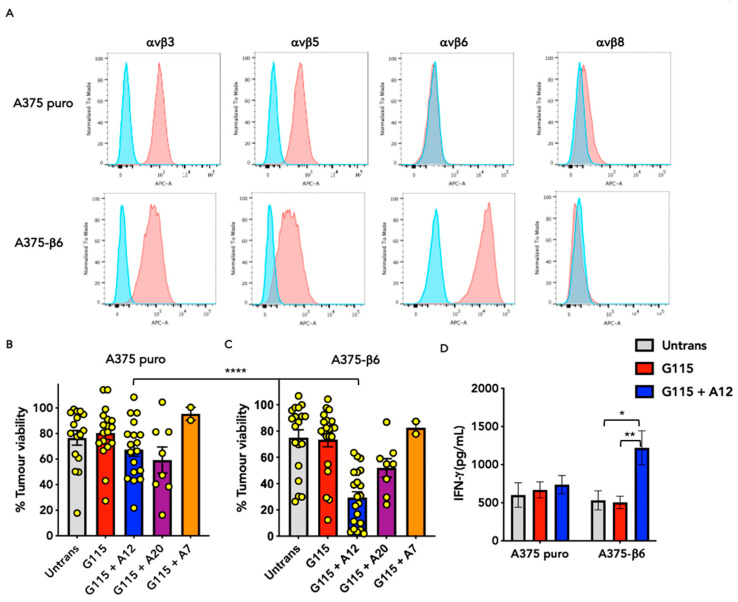
Evaluation of αvβ6-dependent activation of chimeric G115 γδ TCRs. (**A**) Analysis of integrin expression on A375 puro and A375-β6 cells. Red–integrin; blue–isotype control. Data are representative of 3 independent replicates. Co-cultures were performed between A375 puro (**B**) and A375-β6 (**C**) tumour cells and untransduced (untrans) or transduced T-cell populations at an effector-to-target ratio of 1:1 for 72 h. Residual tumour cell viability was determined using an MTT assay (mean ± SEM of indicated replicates). Statistical analysis was performed using one-way ANOVA; **** *p* < 0.0001. (**D**) Supernatants collected from co-cultures described in (**B**,**C**) were analysed for IFN-γ by ELISA (mean ± SEM; n = 21–45 from 4–6 independent donors). Statistical analysis was performed using one-way ANOVA; * *p* < 0.05; ** *p* < 0.01.

**Figure 4 biology-13-00196-f004:**
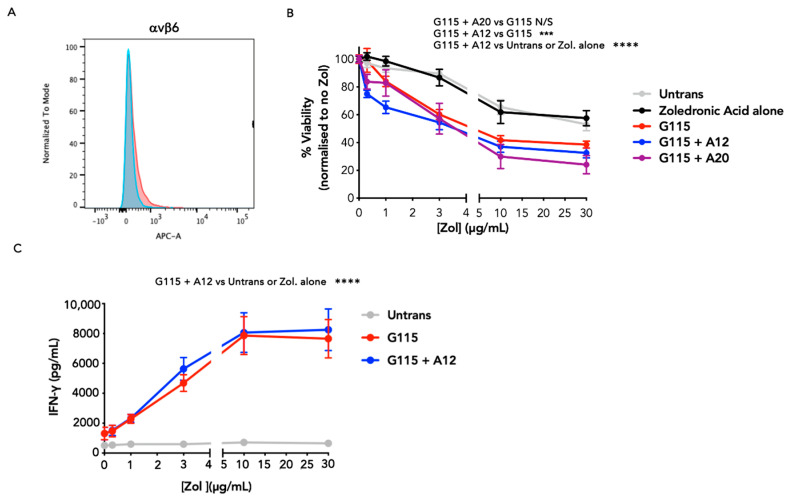
Evaluation of PAg-dependent activation of chimeric G115 γδ TCRs by ffLuc-expressing K562 cells. (**A**) Analysis of αvβ6 integrin expression on ffLuc^+^ K562 cells. Red–integrin; blue–isotype control. Data are representative of 3 independent replicates. (**B**) Firefly luciferase-expressing K562 cells were pre-incubated with the indicated Zol concentration for 24 h prior to the establishment of co-cultures with untrans(duced) or transduced T-cell populations at an effector to target ratio 1:1 for 72 h. Data show mean ± SEM of residual K562 viability (n = 6–11 from 4 independent donors), as determined by luciferase assay. Statistical analysis was performed using two-way ANOVA; *** *p* < 0.001; **** *p* < 0.0001. (**C**) Supernatants collected from co-cultures described in B were analysed for IFN-γ by ELISA (mean ± SEM; n = 19 from 3 independent donors). Statistical analysis was performed using two-way ANOVA; *** *p* < 0.001; **** *p* < 0.0001.

**Figure 5 biology-13-00196-f005:**
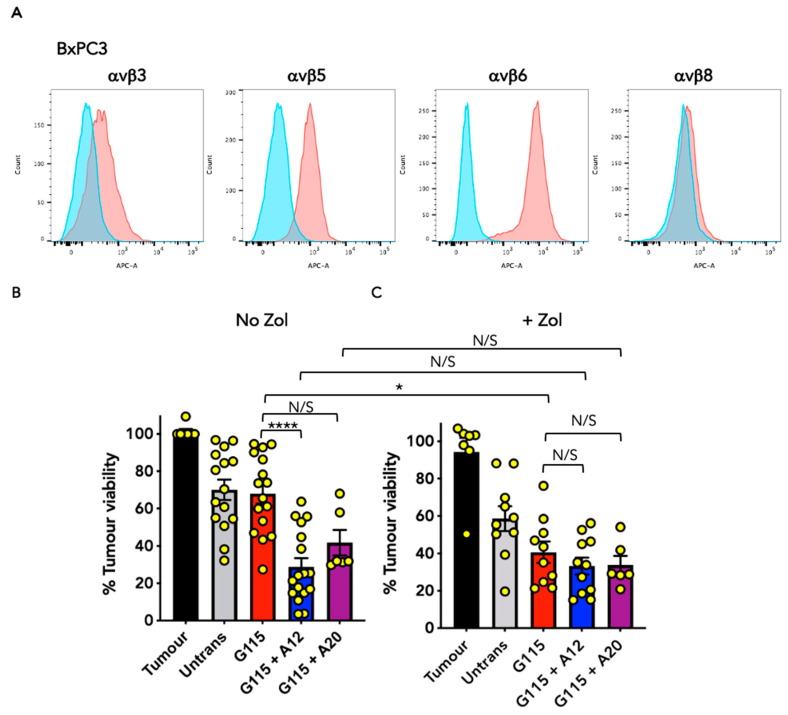
Evaluation of the anti-tumour activity of chimeric G115 γδ TCRs against BxPC3 pancreatic tumour cells. (**A**) Analysis of integrin expression on BxPC3 cells. Red–integrin; blue–isotype control. Data are representative of 3 independent replicates. Co-cultures were performed between BxPC3 tumour cells (No Zol; **B**) or Zol-sensitised BxPC3 tumour cells (+Zol; **C**) and untrans(duced) or transduced T-cell populations at an effector to target ratio of 1:1 for 72 h. Tumour cell viability was determined using an MTT assay (mean ± SEM of indicated replicates). Statistical analysis was performed using one-way ANOVA; **** *p* < 0.0001, * *p* < 0.05, N/S–not significant.

**Figure 6 biology-13-00196-f006:**
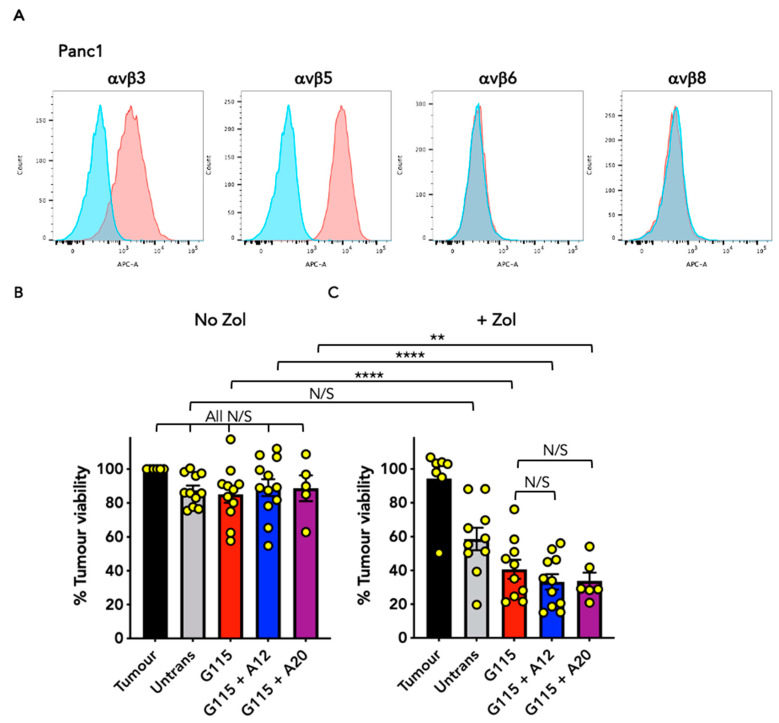
Evaluation of the anti-tumour activity of chimeric G115 γδ TCRs against Panc1 pancreatic tumour cells. (**A**) Analysis of integrin expression on Panc1 cells. Red–integrin; blue–isotype control. Data are representative of 3 independent replicates. Co-cultures were performed between Panc1 tumour cells (No Zol; **B**) or Zol-sensitised Panc1 tumour cells (+ Zol; **C**) and untransduced (untrans) or transduced T-cell populations at an effector to target ratio of 1:1 for 72 h. Tumour cell viability was determined using an MTT assay (mean ± SEM of indicated replicates). Statistical analysis was performed using one-way ANOVA; **** *p* < 0.0001, ** *p* < 0.01, N/S–not significant.

## Data Availability

Data are available from the corresponding author upon request.

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
