# Peer review of "Engineering a Dual Specificity γδ T-Cell Receptor for Cancer Immunotherapy"

_biology, 2024, doi:10.3390/biology13030196_

Round 1

Reviewer 1 Report

Comments and Suggestions for Authors

Authors in present study designed dual specific gd TCRs and introduced this TCR into conventional ab T-cells to broaden the tumor recognition capacity. In addition to natural phosphoantigens recognition capacity of gdTCR, authors introduced foot and mouth virus A20 peptide which binds specifically to avb6 integrin to enable gd TCR to recognize avb6 integrin positive tumors which is an interesting approach to widen the tumor recognition capacity of CART-cells.

The characteristic of MHC independent activation of  gd T-cells make them suitable for allogenic or off shelf T-cell therapy. What is the reason for introducing  gd TCRs into ab T-cells? Instead, authors could have used gdT-cells.

Authors demonstrated that G115+A12 T-cells show superior efficacy in all vitro settings. But there is no in vivo data corroborating invitro data. It would have been better to show some invivo efficacy data.

Author Response

Please see the attachment which addresses all reviewers comments

Reviewer 2 Report

Comments and Suggestions for Authors

In the presented study, the authors highlight the crucial role of γδ T-cells in cancer immunosurveillance, bridging innate and adaptive immunity. The G115 γδ T-cell receptor (TCR) of the Vγ9Vδ2 subtype exhibits responsiveness to phosphoantigens (PAgs) and holds potential for cancer immunotherapy when introduced into conventional αβ T-cells. However, to broaden the cancer specificity of the G115 TCR, the authors aimed to enhance its targeting capabilities by incorporating a tumor-binding peptide (drawn from foot and mouth disease virus) into the complementarity determining region (CDR) 3 of the TCR δ2 chain. These engineered T-cells demonstrated potent cytotoxicity against both PAg-presenting tumor cells and those expressing αvβ6 integrin.

It is a well-conducted study, and I suggest a revision of the manuscript to increase readership:

Novelty: The authors have mentioned that such studies have been done previously, thus questioning the novelty of their research. I would suggest they direct comparisons with such studies in the discussion and emphasize how their study takes the community forward.

Transduction Efficiency: The successful transduction of T-cells with the SFG retroviral vector encoding the G115 γδ TCR is essential for subsequent experiments. It's noted that transduction efficiency was not hindered by incorporating different peptides, indicating a robust transduction process. However, please discuss the exact transduction efficiency and quantify it to ensure consistency and reliability across experiments.

Pan antibodies: Were TCR-specific antibodies or tetramer staining tried?

Competition with Endogenous TCR: It's noted that expression of the endogenous αβ TCR was reduced, but not abolished, in transduced T-cells due to competition for endogenous CD3 subunits. This observation highlights the potential for interference between the engineered γδ TCR and endogenous TCR signaling pathways, which could impact the functional behavior of the transduced T-cells. Please elaborate.

Generalizability of Findings: The study extended its evaluation to pancreatic tumor cells (BxPC3 and Panc1) to assess the generality of the observed effects. While the results with BxPC3 cells support the notion of αvβ6 engagement and enhanced cytolytic activity with G115 + A12 T-cells, the interpretation of results with Panc1 cells is complicated by their lack of αvβ6 integrin expression. Further experiments, such as integrin expression analysis or functional validation, are needed to clarify the role of αvβ6 integrin in the observed cytotoxicity.

Plan for in-vivo studies: There is a need for deeper mechanistic insights into the observed cytotoxicity and cytokine release, such as elucidating the signaling pathways involved in TCR activation and effector function. Moreover, in vivo studies are crucial to validate the therapeutic potential of these chimeric γδ TCRs in relevant preclinical models of cancer. Do the authors plan for this?

Reviewer 3 Report

Comments and Suggestions for Authors

The paper provides a comprehensive exploration of engineering a dual-specificity γδ T-cell receptor for cancer immunotherapy. The paper introduces the background information on γδ T-cells, emphasizing their significance in cancer immunosurveillance and the limitations of current immunotherapy strategies utilizing these cells.

The materials and methods section is well-detailed, providing clarity on the experimental procedures, cell lines, and constructs used in the study. The inclusion of figures, such as the engineered TCR sequences and vector structures, aids in understanding the experimental design.

The results presented are robust, demonstrating successful engineering of the G115 TCR with the A12 peptide, leading to a dual-specificity TCR capable of recognizing both PAgs and αvβ6 integrin. The authors appropriately use flow cytometry and cytotoxicity assays to validate TCR expression and functionality.

Furthermore, the study examines the dual-specificity TCR's effectiveness against different cancer cell lines, including those expressing αvβ6 integrin and those responding to phosphoantigens. The inclusion of additional figures for each cell line strengthens the comprehensiveness of the results.

The conclusion appropriately summarizes the key findings, emphasizing the successful creation of a dual-specificity TCR with potential implications for cancer immunotherapy. 

The discussion section delves into the broader context of the study, addressing the significance of dual-specificity in T-cell engineering and potential avenues for future research. I  recommended: 

  • Expand the discussion section to include a comparison with existing literature and potential implications of the findings.
  • Provide insights into the significance of the study's outcomes for cancer immunotherapy.

Author Response

Please see attachment in response to reviewer 1 which addresses all reviewer comments
